# Predictors of Satisfaction with Care Services among Family Members of Older Adult Residents of Long-Term Care Facilities

**DOI:** 10.3390/ijerph17093298

**Published:** 2020-05-09

**Authors:** Eun-Ok Song, Hye-Young Jang

**Affiliations:** College of Nursing, Hanyang University, Seoul 04763, Korea; wkqjr100@hanyang.ac.kr

**Keywords:** satisfaction, care, family members, long-term care facilities

## Abstract

This study identified predictors of satisfaction with care services among family members of older adults residing in long-term care facilities (LTCFs). In this cross-sectional descriptive study, the participants were 330 family members of older adult residents of LTCFs in Seoul, Gyeonggi, Gangwon, Gyeongbuk, and Chungnam, Korea. Data were collected from July to October 2018 using a structured self-report questionnaire. Data were analyzed using descriptive statistics, independent t-testing, one-way ANOVA, Kruskal–Wallis testing, Pearson’s correlation coefficients, and hierarchical multiple regressions. The most important predictors of satisfaction with care services were satisfaction with the physical housing environment (β = 0.49, *p* < 0.001), caregiving stress (β = −0.30, *p* < 0.001), the facility’s size (β = −0.13, *p* = 0.001), the number of visits to the facility (β = −0.10, *p* = 0.024), and the number of family members who participated in the decision to place the relative in a facility (β = 0.09, *p* = 0.033). This study is significant because it provides fundamental data for qualitatively improving care services in LTCFs. Based on the results, strategies should be developed to relieve caregiving stress among family members and improve satisfaction with the physical housing environment.

## 1. Introduction

Due to an increase in average life expectancy and decline in birthrates, the older adult population has been growing rapidly on a worldwide scale. Korea is no exception; 14.3% of all inhabitants were 65 years or older as of 2018, and it is predicted that 41% of Koreans will be senior citizens by 2060 [1]. This increase in the elderly population has led to an increase in cases of geriatric disorders such as dementia, Parkinson’s disease, and strokes and has brought about various social problems related to caregiving for older adults, who often require long-term care [2]. Family members complain of physical, mental, social, and economic stress caused by caregiving [2]. Caregiving stress has induced an expanding number of older adults to enter long-term care facilities (LTCFs). As of 2016 in Korea, there were 345,000 residents in such facilities [3].

Family members anticipate less caregiving stress for themselves and a better quality of life for their loved ones who enter an LTCF. They choose facilities based on diverse factors, including the setting and services provided [4,5]. Because facilities’ primary function is to protect and care for older adults in place of family members [6], family members expect LTCFs to respect their loved ones’ rights and provide them with comfortable, professional care so that they can live sound, healthy lives within a community. Such expectations are reflected through the satisfaction levels family members report for facilities’ services after experiencing them.

User satisfaction is the subjective perception by service users that service providers and institutions should accept as reality given that it is a practical reflection of care and quality of care [7]. Indeed, satisfaction surveys are a major method used to evaluate the quality of care services from the user’s point of view because the survey results reflect the extent to which the provided nursing service meets the various expectations of the user [8]. Investigations into satisfaction not only identify individuals’ complex feelings but also create an important starting point when looking for ways to enhance institutions’ services [9]. Thus, it is necessary to assess user satisfaction with facilities to provide a foundation for strengthening the services provided.

On the one hand, there are many challenges associated with conducting satisfaction surveys among LTCF residents; these residents often have cognitive deficits, low response rates [10], and lower expectations as they adapt to their environment [11]. On the other hand, because family members can understand their older family member’s condition well and evaluate objective service aspects from an external perspective [12], there is increased interest in measuring satisfaction from the perspective of the family members. Previous studies have reported that family members’ satisfaction is more consistent with existing quality measures than that of residents [12,13]. In addition, obtaining valid information from multiple stakeholders is important to obtain an understanding of the different perspectives of the residents [13], and in this respect, family members have different perspectives of LTCF quality than residents. Family members are typically more critical of LTCF care than residents are [14], as family members can maintain a more external perspective than residents [11]. Therefore, it is necessary to assess facility service satisfaction from the family’s perspective. Family satisfaction with facility services can be seen as an evaluation of the quality of service and can be used to improve the quality of service.

In Western countries, previous studies on satisfaction with the services provided by LTCFs have reported that residential environment, costs paid by users, services, and the professionalism of the staff are key factors affecting satisfaction [15,16,17,18]. In addition, a smaller size, nonprofit ownership, rural location, lower cost, and higher staff expertise were associated with greater family satisfaction [15,16,17,18]. In Korea, satisfaction with general lifestyle management services (such as eating, housework, and bathing) provided to older adults residents is high, whereas satisfaction with services that require professionalism (such as social participation and medical services) is poor [19]. Satisfaction is high when facilities have positive awareness and attitudes regarding the quality of their services. However, the multidimensional components of service settings tend to affect user satisfaction because of their subjective features [17,20]. Therefore, it is important to explore the factors that influence satisfaction with the services offered by LTCFs from a multidimensional perspective.

We conducted this study to identify predictors of satisfaction with care services among family members of older adult residents in LTCFs using a multilevel framework [17]. In this framework, family satisfaction can be influenced by family characteristics, the relationship between the family and resident, and the facility characteristics. In this study, we evaluated variables identified as pertinent in previous studies [17,20].

The characteristics of the family included the number of visits to the facility, duration of caregiving at home, number of individuals that participated in the decision for the relative to be placed in a facility, participation in direct care, and relationships with the resident. Among the factors known to affect satisfaction, caregiving stress experienced by family members is a major variable that reflects users’ subjectivity and is deeply linked to satisfaction with care services [21]. Although it might be expected that family members experience less stress after their loved one has entered an LTCF, many family members still feel stress due to a sense of guilt or internal conflict about sending their loved one to live in a facility, conflicts with facility staff when they participate in caregiving [22], and a sense of loss [4,5,21,23,24]. Thus, caregiving stress among family members should be considered as a factor that affects satisfaction with services.

Facility characteristics that we evaluated included facility size and physical housing environment. Facilities should provide physical environments for residents to maintain their residual functions and allow them to live as independently as possible. The physical housing atmosphere is a critical element that affects the daily activities of residents with poor physical function [25]; unsanitary, poor housing settings negatively affect both quality of life and health [26]. Therefore, the housing environment should be taken into account when choosing a care facility because it influences satisfaction with the facility’s services [5,27]. Satisfaction with care services comprises all opinions related to how the needs and desires of users are met; it is determined by both service content and the appropriateness of the surrounding context. Therefore, it is necessary to consider the physical housing environment as a factor influencing satisfaction.

This study identifies predictors of satisfaction with care services among family members of older adult residents in LTCFs. Thus, it provides basic data that LTCFs can use to improve the quality of their care services and contributes to the development of paths for improvement.

## 2. Methods

### 2.1. Study Design and Setting

This study used a cross-sectional design to identify factors predictive of satisfaction with care services among family members of older adult residents in LTCFs. Participants were recruited using convenience sampling from 58 LTCFs in Seoul, Gyeonggi, Gangwon, Gyeongbuk, and Chungnam, Korea. Inclusion criteria were as follows: family members who visited their relatives frequently (out of all family members), played a major role as caregivers, understood the study’s purpose, and gave consent to participate.

We chose to recruit participants from 58 facilities (160 participants in 50 facilities with less than 100 beds, and 170 participants in 8 facilities with more than 100 beds) based on the current status of LTCFs in Korea in 2017 (5014 facilities with less than 100 beds, 222 facilities with more than 100 beds) [28] and the distribution of facilities in Seoul, Gyeonggi, Gangwon, Gyeongbuk, and Chungnam. The study’s purpose was explained in person to the heads of the facilities and permission to recruit was requested. Upon approval from the heads of the facilities, managers in the facilities provided a list of residents’ family members who were potentially eligible to participate in the study. We then contacted each family member independently and explained the purpose and process of this study.

A total of 360 questionnaires were distributed, of which 336 questionnaires were returned (response rate: 91.7%); six were excluded because the respondents did not include demographic characteristics (e.g., age, education level). We used the G*Power 3.1.9.2 program to calculate the number of participants required. The sample size required for multiple regression analysis was 217, with an effect size of 0.15, a significance level of 0.05, power of 0.95, and 19 predictors [29]. By recruiting 330 family members, we met the minimum sample population requirement.

### 2.2. Measures

To identify predictors of satisfaction with care services, we used independent variables based on a multilevel framework [17].

#### 2.2.1. Demographics and Facility Care-Related Characteristics

Participant general characteristics included gender, age, education level, marital status, occupation, perceived economic status, perceived health status, perceived stress, relationship with the resident, number of chronic diseases, and duration of caregiving at home before the resident entered the facility. General characteristics of the residents that were collected were gender, age, and diagnosis. For facility care-related characteristics, the size of the facility, number of visits to the facility, duration of each visit, number of family members who participated in the decision to place the relative in a facility, identity of the primary family member who made the decision to place the relative in a facility, who paid for the facility, and participation in direct care during visits (going on walks, assisting in ambulation and eating, hair grooming and other kinds of grooming, playing and engaging in conversations, etc.) were collected. The duration of caregiving at home, duration of the visits, and number of visits per month were classified as described in previous studies [21,30].

#### 2.2.2. Caregiving Stress

Participant caregiving stress was assessed using the Korean version [30] of the Family Perceptions of Caregiver Roles developed by Mass and Buckwalter [31]. The subcategories contain 32 items about conflicts with facility staff (12 items; e.g., “I feel like an outsider in the care of my relative.”), internal conflict (8 items; e.g., “I feel that my health has suffered because of my involvement in care.”), sense of guilt (5 items; e.g., “I feel that I don’t do as much for my loved one in the facility as I could or should.”), and sense of loss (7 items; e.g., “Loss of meaningful interaction with patient.”). Responses were based on a 5-point Likert scale: 1 point for “not at all”, 2 for “slightly”, 3 for “normal”, 4 for “strongly”, and 5 for “very strongly”. Possible scores thus ranged from 32 to 160 points with a higher score indicating greater caregiving stress. The study by Park et al. [30] had a Cronbach’s α of 0.90; the Cronbach’s α of this study was also 0.90.

#### 2.2.3. Satisfaction with Physical Housing Environment

As a tool to measure satisfaction with the physical housing environment, overall satisfaction was measured using an item-categorized visual analog scale ranging from 0 to 10 based on previous studies that measured overall satisfaction on a 10-point scale [32,33]. Participants were asked “Overall, how satisfied are you with the housing environment of the long-term care facility?” Participants responded using a numerical rating scale that ranged from 0 for “highly dissatisfied” to 10 for “highly satisfied”. A greater score thus signified more satisfaction with the physical housing environment.

#### 2.2.4. Satisfaction with Care Services

Satisfaction with care services was evaluated using the Korean version [30] of the Family Perceptions of Care Tools developed by Mass and Buckwalter [31]. Subcategories comprised 25 items about consideration by staff (7 items; e.g., “Staff provide for the privacy of my family member.”), efficient management (7 items; e.g., “If more resources were available, staff could provide care that would be more beneficial for my family member.”), physical nursing care (6 items; e.g., “Grooming and hygiene”), and activities (5 items; e.g., “I am satisfied that there are sufficient opportunities for my family member to enjoy the outdoors and other diversions.”). The responses were based on a 5-point Likert scale: 1 point for “strongly disagree”, 2 for “disagree”, 3 for “normal”, 4 for “agree”, and 5 for “strongly agree”. The possible range of scores was 25–125 points and a higher score indicated more satisfaction with care services. Cronbach’s α was 0.94 in the study by Park et al. [30] and 0.88 in this study.

### 2.3. Data Collection and Ethical Considerations

Data were collected after obtaining approval from the Institutional Review Board (IRB no. HYI-17-086-1) of Hanyang University. Data were collected from July to October 2018. We visited the LTCFs to explain the study’s purpose to the heads of these facilities and ask for permission to recruit. Once this permission was provided, we explained the study’s purpose and process to the participants, along with their rights, voluntary participation, and confidentiality. Only those who volunteered to take part were asked to fill out the questionnaire after giving written consent.

### 2.4. Statistical Analysis

Data were analyzed using the SPSS/WIN program version 25.0 (IBM Corp, Armonk, NY, USA). Descriptive analyses of all general characteristics, facility care-related characteristics, and main variables were conducted. To explore differences in satisfaction with care services according to general characteristics and facility care-related characteristics, independent t-tests, one-way analysis of variance (ANOVA), and a post hoc Scheffe test were used for variables that followed a normal distribution after the normality test, whereas for the variables that did not follow a normal distribution, the Kruskal–Wallis test was used. Correlations among caregiving stress, satisfaction with the physical housing environment, and satisfaction with care services were evaluated using Pearson correlation coefficients. Hierarchical multiple regressions were conducted to identify predictors of satisfaction with care services. All *p* values < 0.05 were considered to indicate statistical significance. The power of this study was 0.99 based on an effect size (f^2^) of 0.25, sample size of 330, error probability (∝) of 0.05, and two-tailed testing [29].

## 3. Results

### 3.1. Characteristics of the Family Members

In this study, 336 questionnaires (response rate: 91.7%) were returned after the distribution of 360 copies. Three hundred and thirty questionnaires were included in the final analysis after excluding six questionnaires with incomplete responses. Most participants were female (62.1%), the average age was 53.67 ± 11.04 years, 221 (67.0%) were college graduates, and 299 (90.3%) were married. Most participants were unemployed (59.7%), ranked in the middle for perceived economic status (80.3%), and ranked themselves in good health with regard to perceived health status (45.8%). Participants responded that they sometimes felt stress (77.6%), and the majority relationship type with the resident was adult child (60%). Participants had an average of 1.35 ± 0.67 chronic diseases and the average duration of caregiving at home prior to placement of the relative in a LTCF was 53.52 ± 89.85 months (Table 1).

### 3.2. Characteristics of the Residents at LTCFs

Of the residents, 248 (75.2%) were women, and the average age was 83.94 ± 8.34 years. The most common diagnosis among the residents was dementia (45.6%), followed by cerebrovascular disease (22.5%), arthropathy (14.4%), Parkinson’s disease (7.9%), and cancer (1.3%) (Table 2).

### 3.3. Facilities’ Care-Related Characteristics

The most common facility size was 100 beds or more (51.5%), and 264 (80.0%) participants visited less than four times a month, with 245 (74.3%) staying for less than two hours per visit. The number of family members who participated in the decision to place the relative in a facility was usually more than two (54.8%), with the primary decision-maker being a son of the resident (34.7%), and sons generally paid for the facility (40.4%). Most participants provided direct care (75.2%) (Table 3).

### 3.4. Descriptive Statistics of Measured Variables

Descriptive statistics for the study variables are presented in Table 4. The average caregiving stress score among participants was 83.61 ± 17.22 (range = 37–128). On average, satisfaction with the physical housing environment was 8.02 ± 1.72, and satisfaction with care services was 86.92 ± 12.52.

### 3.5. Differences in Satisfaction with Care Services According to Demographics and Facility Care-Related Characteristics

Table 5 shows the differences in satisfaction with care services according to participant demographics and facility care-related characteristics. Satisfaction with care services differed significantly according to participant gender (t = −1.97, *p* = 0.050), perceived stress (χ^2^ = 7.17, *p* = 0.028), relationship with the resident (χ^2^ = 12.64, *p* = 0.013), duration of caregiving at home (F = 4.72, *p* = 0.010), size of the facility (t = 4.67, *p* < 0.001), number of monthly visits to the facility (F = 10.77, *p* < 0.001), the duration of each visit (χ^2^ = 7.03, *p* = 0.030), the number of decision-makers (t = −2.55, *p* = 0.011), and the direct care the participant themselves provided (t = −2.96, *p* = 0.003) (Table 5).

### 3.6. Correlations between Measured Variables

Participant satisfaction with care services was significantly negatively correlated with caregiving stress (r = −0.44, *p* < 0.001), conflicts with facility staff (r = −0.50, *p* < 0.001), internal conflict (r = −0.39, *p* < 0.001), guilt (r = −0.13, *p* = 0.017), and loss (r = −0.16, *p* = 0.005). By contrast, satisfaction with care services and satisfaction with the physical housing environment had a significant positive correlation (r = 0.62, *p* < 0.001) (Table 6).

### 3.7. Predictors of Family Caregivers’ Satisfaction with Care Services in LTCFs

Before the analysis, the presence of multicollinearity was tested, and the correlation coefficient among independent variables was less than 0.6, the tolerance of all variables was larger than 0.1, and the variance inflation factor was less than 10. Because the Durbin–Watson value was 1.89, which is close to 2, residuals were considered to be independent [34,35].

Table 7 summarizes the results of the hierarchical multiple regression model for factors associated with satisfaction with care services. Stage 1 of the hierarchical regression analysis took into account gender, perceived stress, relationship with the resident, duration of caregiving at home, number of monthly visits to the facility, duration of each visit, identity of the person who made the decision to place the relative in a facility, number of family members who made the decision to place the relative in a facility, and whether or not the family member provided direct care. In the first model (adjusted R^2^ = 0.17, F = 6.13, *p* < 0.001), predictive factors were the number of visits (β = −0.19, *p* = 0.001), size of the facility (β = −0.19, *p* = 0.001), duration of caregiving at home (β = 0.13, *p* = 0.013), providing direct care (β = −0.12, *p* = 0.025), and number of family makers who made the decision to place the relative in a facility (β = 0.11, *p* = 0.043). The final model’s explanatory power increased by 34% (R^2^ change = 0.34) after adding caregiving stress and satisfaction with the physical housing environment. In the final model (adjusted R^2^ = 0.52, F = 24.26, *p* < 0.001), better satisfaction with the physical housing environment (β = 0.49, *p* < 0.001) was the strongest factor predictive of satisfaction with care services, followed by less caregiving stress (β = −0.30, *p* < 0.001), smaller facility (β = −0.13, *p* = 0.001), higher number of monthly visits to the facility (β = −0.10, *p* = 0.024), and a larger number of decision-makers who made the decision to place the relative in a facility (β = 0.09, *p* = 0.033).

## 4. Discussion

This study was conducted to explore the effects of caregiving stress and satisfaction with the physical housing environment and care services among family members of older adult residents in LTCFs on their overall satisfaction with the care services.

We conducted hierarchical regression analysis to identify predictors of satisfaction with care services among family members of relatives in LTCFs. In stage 1, predictors were the number of visits, size of the facility, duration of caregiving at home, providing direct care, and number of family members who made the decision to place the relative in facility; the explanatory power of this model was 17%.

The most crucial predictor was the number of visits to the facility; a lower number of visits was associated with higher satisfaction with care services. This result differs from those reported in previous studies. Shippee et al. [17] reported that family members who visited more than once a week were more satisfied with facilities than those who visited less than once a week. Tsai, Tsai, and Huang [36] also found that family members who visited more frequently were more satisfied with facilities. Visits from family members correlate with improved well-being among residents [37,38]. Family members maintain their relationships with residents by visiting them and giving feedback staff in terms of the care provided and its quality [39]. On the other hand, there is a view that excessively frequent visits are a sign of dissatisfaction with the facility’s care services. Nijkamp et al. [32] reported that when families are dissatisfied with care, they more frequently visit a facility to monitor the provision of care for the resident. In addition, previous studies have shown that the number of visits is affected by diverse factors. In other words, the number of visits could decrease as the duration of a resident’s stay in a facility increases [40], when family members become more emotionally difficult to deal with, when a family member’s sense of guilt declines, and/or when needs increase in other areas of life [41]. Therefore, for more consistent research results, it is necessary to identify not only the number of visits but also factors affecting the number of visits, including the purpose of visits.

In addition, satisfaction with care services was higher in facilities with less than 100 beds than in those with 100 beds or more. This finding is similar to those of previous studies, in which family members were more satisfied with smaller facilities [15,16]. In smaller facilities, facility staff have a relatively broader range of duties and responsibilities than in larger facilities [42], which leads to frequent interactions with family members; more frequent interactions could explain why satisfaction with care services was higher for smaller facilities.

In addition, the longer the duration of caregiving at home, the higher the family member’s satisfaction was with the LTCF. This finding can be explained by the fact that the burden of caregiving increases with the duration of the care period [5], hence those family members who had been caring for relatives longer likely experienced relatively more alleviation of their caregiving stress than those who had been caring for a shorter period for their relative, leading to greater satisfaction with care services. Furthermore, the less the family member participated directly in care for the relative in the facility, the higher their satisfaction with the facility. In other words, family members who participated in direct caregiving (such as by going on walks and assisting with ambulation, eating, and grooming) were less satisfied with care services than those who did not take part in such activities, which is consistent with the results of previous studies, which found that family members did not wish to increase their participation in direct caregiving [16,43,44]. In agreement with our finding, another research study also found that family members who took part in more daily activities (such as eating and bathing their relative) were less satisfied with the facility’s caregiving; however, those family members who frequently communicated with staff were more satisfied with the facility’s caregiving [45]. Those results imply that the level of satisfaction varies depending on the type of participation in caregiving at the facility. Thus, future studies should attempt to link satisfaction with care services with the type of family member participation with the ultimate goal of developing a program for family participation in the care of a relative in an LTCF.

In stage 2, the most critical predictors of satisfaction with care services were satisfaction with the physical housing environment and caregiving stress. Housing environment plays a critical role in family decision-making regarding whether to send a loved older relative to a facility [27]. In addition, the physical housing environment affects the health and well-being of residents with poor physical function [46], such that satisfaction with the environment affects satisfaction with care services. To enhance satisfaction with care services, facilities need to identify and solve problems related to the physical housing environment. Furthermore, staff should identify family members’ needs through continual interactions with them and try to decrease their remaining stress, which continues to exist even after their loved ones have become LTCF residents. Specific measures to reduce a family’s caregiving stress include identifying the family’s unmet needs through regular consultations, sharing information and decision-making on facility care to offset the guilt or loss experienced from placing the relative in a facility, and developing family care participation programs to enable participation.

In this study, the average caregiving stress score among family members was 83.61 out of 160. This finding is similar to Im et al.’s findings [21]; these authors reported a moderate amount of caregiving stress among family members of dementia patients in daycare centers and nursing homes. Thus, family caregivers still feel stress even after they transfer their loved one to a care facility, possibly through a sense of guilt or internal conflict, conflicts with facility staff when participating in care [22], or a sense of loss [21]. Caregiving stress affects not only the health of family members, but also the health and quality of life of residents [47]. Roberts and Ishler [45] suggested that to decrease caregiving stress, family members should participate in care by interacting with residents, providing direct care, and interacting with facility staff during visits. Satisfaction with the physical housing environment was high, an average of 8.02 out of 10. The physical housing environment affects the daily activities of residents with poor physical function [25], and poor housing settings negatively affect residents’ quality of life and health [26]. In addition, the physical housing environment should be considered when family caregivers make the choice to send and keep their loved one at a care facility [5,27]. Therefore, the heads and staff of LTCFs should make continuous efforts to ensure a safe, sanitary, comfortable, and familiar environment [46]. However, it is important to note that we measured satisfaction with the physical housing environment using only one item, which may have reduced the sensitivity of this measurement. Future studies should assess satisfaction with the physical housing environment by looking at the subcategories of safety, health, and convenience, among others. In addition, there are likely to be differences in the basic residential environment depending on the size of the LTCF, therefore future studies need to identify differences in family satisfaction with the residential environment according to the size of the facility to prepare a more individualized residential environment improvement plan.

In this study, the average satisfaction with care services was 82.92 out of 125, corresponding to a moderate level of satisfaction. Because most residents stay in facilities for the rest of their lives, it is critical for them to receive quality care services to ensure that they have healthy, enjoyable experiences. In the present study, satisfaction with care services was higher among the children of residents than among the spouses of residents. This finding is consistent with Shippee et al.’s study [17]; these researchers demonstrated that children of the relative in a facility were more satisfied with facilities than spouses were. Spouses of individuals admitted to LTCFs are likely also experiencing physical aging and deterioration, may have chronic diseases, and be experiencing social status changes and loss of their role of spouse, thereby increasing their daily stress [48], which could explain why there were less satisfied with care services than the adult children of the LTCF resident. Because the relationship between the family caregiver and the resident affects satisfaction with care services [49], it is necessary to identify specific causes of different levels of satisfaction with care services in future studies.

In this study, family caregivers who provided care for their loved ones for at least two years before entrusting them to a facility had significantly higher satisfaction with care services than those who provided care for less than two years. That finding is similar to Tornatore and Grant’s findings [16] that family members who provided care for a longer time before sending their loved ones to a facility were more satisfied with the facility. In the present study, family members who visited for 2–3 h at a time were less satisfied with care services than those who stayed for less than two hours or more than three hours. Family members who stay for a long time (more than three hours) have more opportunities to participate in caregiving through interactions with staff and other residents, which could lead to higher satisfaction with the facilities. McVeigh et al. [50] reported that increased interactions with staff increased satisfaction with care services, consistent with our findings. Moreover, satisfaction with care services was higher when two or more family members participated in the decision to place their loved one in a facility than when only one family member made the choice. Perhaps more factors are considered when two or more family members participate in the decision, leading to greater satisfaction with the care services [50]. Further research based on qualitative in-depth interviews should explore how the decision to place a family member in a facility is made by the family.

A strength of this study was that satisfaction with care services was assessed from the perspective of families, who are important decision-makers in facility care. Family members likely have stricter standards regarding care than the resident of the LTCF himself or herself for various reasons, including cognitive decline in the resident and the resident’s familiarity with the environment [14]. It is important to obtain valid information from multiple stakeholders regarding the LTCF environment and quality of care services to consider various viewpoints [13].

Limitations of this study are as follows. First, as noted previously, satisfaction with the physical housing environment was measured by a single question. More detailed measures should be used in future studies. Second, the reasons why the participating family member visited his or her relative were not directly verified. In future studies, the purpose of the visit should be identified to assess the relationship between the number of visits and satisfaction more accurately. Lastly, this study was conducted only in specific areas, thus caution should be used when generalizing the results.

## 5. Conclusions

Family caregivers who are satisfied with care services at an LTCF can provide positive feedback to other families who are contemplating sending their loved ones to such homes and help them make decisions that align with their values [50]. In this study, we identified predictors of family members’ satisfaction with care services, which can be used as basic data to qualitatively improve services at LTCFs. We found that the most critical predictors of satisfaction with the care provided at the LTCFs were the physical housing environment and caregiving stress. To improve satisfaction with the physical housing environment, careful inspections and regular evaluations of cleanliness, pleasantness, comfort, and safety should be instituted [17,27,46]. To decrease caregiving stress, strategies should be developed to encourage and help family members to participate in caregiving, which will decrease their sense of guilt, internal conflict, or loss after sending their loved one to a facility. To build trust with staff, an approach that fosters and maintains constructive staff–family member relationships is needed, especially when family members participate in caregiving.

## Figures and Tables

**Table 1 ijerph-17-03298-t001:** Characteristics of the family members, *N* = 330.

Characteristics	Category	*n* (%)	Mean ± SD (Range)
Gender	Female	205 (62.1)	
	Male	125 (37.9)	
Age (years)			53.67 ± 11.04 (18–88)
	≤39	33 (10.0)	
	40–49	67 (20.3)	
	50–59	136 (41.2)	
	≥60	94 (28.5)	
Education level	≤High school	109 (33.0)	
	≥College	221 (67.0)	
Marital status	Single	31 (9.4)	
	Married	299 (90.6)	
Occupation	Employed	133 (40.3)	
	Unemployed	197 (59.7)	
Perceived economic status	High	28 (8.5)	
	Middle	265 (80.3)	
	Low	37 (11.2)	
Perceived health status	Fair	43 (13.0)	
	Good	151 (45.8)	
	Poor	136 (41.2)	
Perceived stress	Very often	16 (4.8)	
	Sometimes	256 (77.6)	
	Almost never	58 (17.6)	
Relationship with resident	Spouse	16 (4.8)	
	Adult child	198 (60.0)	
	Daughter-in-law	59 (17.9)	
	Son-in-law	23 (7.0)	
	Other	34 (10.3)	
Number of chronic diseases			1.35 ± 0.67 (0–4)
	0	157 (47.6)	
	1	90 (27.3)	
	≥2	83 (25.1)	
Duration of caregiving at home (months)			53.52 ± 89.85 (0–750)
	0	86 (26.1)	
	1–24	92 (27.9)	
	≥25	152 (46.1)	

SD, standard deviation.

**Table 2 ijerph-17-03298-t002:** Characteristics of the residents at long-term care facilities (LTCFs), *N* = 330.

Characteristics	Category	*n* (%)	Mean ± SD (Range)
Gender	Female	248 (75.2)	
	Male	82 (24.8)	
Age (years)			83.94 ± 8.34 (65–102)
	≤75	45 (13.6)	
	76–85	137 (41.5)	
	≥86	148 (44.9)	
Diagnosis *	Dementia	203 (45.6)	
	Cerebrovascular disease	100 (22.5)	
	Arthropathy	64 (14.4)	
	Parkinson’s disease	35 (7.9)	
	Cancer	6 (1.3)	

SD, standard deviation; * multiple response.

**Table 3 ijerph-17-03298-t003:** Facility care-related characteristics, *N* = 330.

Characteristics	Category	*n* (%)	Mean ± SD (Range)
Size of facility	≤99 beds	160 (48.5)	
	≥100 beds	170 (51.5)	
Number of visits to the facility (per month)			4.77 ± 5.16 (0–30)
	≤4	264 (80.0)	
	5–9	36 (10.9)	
	≥10	30 (9.1)	
Duration of each visit (hours)	≤1	245 (74.3)	
	2–3	71 (21.5)	
	≥4	14 (4.2)	
Number of decision-makers regarding placement of relative in facility			1.93 ± 1.12 (0–8)
	≤1	149 (45.2)	
	≥2	181 (54.8)	
Final decision-maker regarding placement of the relative in a facility *	Son	221 (34.7)	
	Daughter	171 (26.9)	
	Daughter-in-law	69 (10.8)	
	Spouse	64 (10.1)	
	Himself or herself	61 (9.6)	
	Son-in-law	26 (4.1)	
	Relative	11 (1.7)	
	Other	13 (2.1)	
Person who paid for the facility *	Son	197 (40.4)	
	Daughter	130 (26.6)	
	Himself or herself	65 (13.3)	
	Spouse	43 (8.8)	
	Daughter-in-law	20 (4.1)	
	Son-in-law	16 (3.3)	
	Relative	5 (1.0)	
	Other	12 (2.5)	
Family member participating in direct care	Yes	248 (75.2)	
	No	82 (24.8)	

SD, standard deviation; * multiple response.

**Table 4 ijerph-17-03298-t004:** Descriptive statistics for measured variables, *N* = 330.

Variable	Mean ± SD	Observed Range	Possible Range
1. Caregiving stress	83.61 ± 17.22	37–128	32–160
Conflicts with staff	29.28 ± 6.83	12–47	12–60
Captivity	20.40 ± 6.39	8–39	8–40
Guilt	13.31 ± 3.90	5–25	5–25
Loss	20.64 ± 6.47	7–35	7–35
2. Satisfaction with the physical housing environment	8.02 ± 1.72	0–10	0–10
3. Satisfaction with care services	86.92 ± 12.52	39–119	25–125
Consideration	25.77 ± 4.71	7–35	7–35
Management effectiveness	24.40 ± 4.48	11–35	7–35
Physical care	22.33 ± 3.94	8–30	6–30
Activities	14.39 ± 1.97	5–25	5–25

SD, standard deviation.

**Table 5 ijerph-17-03298-t005:** Differences in satisfaction with care services according to demographics and facility care-related characteristics, *N* = 330.

Characteristics	Category	Mean ± SD	χ^2^ or t/F (*p*)
Gender	Female	85.87 ± 13.12	−1.97 (0.050)
	Male	88.66 ± 11.30	
Age (years)	≤39	86.79 ± 16.11	2.31 (0.077)
	40–49	86.22 ± 11.91	
	50–59	88.90 ± 11.10	
	≥60	84.61 ± 13.21	
Education level	≤High school	86.38 ± 11.76	−0.55 (0.584)
	≥College	87.19 ± 12.89	
Marital status	Single	90.29 ± 14.64	1.58 (0.116)
	Married	86.58 ± 12.26	
Occupation	Employed	86.28 ± 13.92	−0.74 (0.459)
	Unemployed	87.36 ± 11.50	
Perceived economic status *	High	88.89 ± 8.86	0.04 (0.977)
	Middle	85.86 ± 12.26	
	Low	87.19 ± 12.90	
Perceived health status	Fair	88.18 ± 11.27	2.53 (0.081)
	Good	85.26 ± 11.74	
	Poor	88.81 ± 17.57	
Perceived stress *	Very Often	78.68 ± 20.68	7.17 (0.028)
	Sometimes	86.74 ± 11.80	
	Almost Never	90.14 ± 11.74	
Relationship with the resident *	Spouse	77.69 ± 16.78	12.64 (0.013)
	Adult child	87.19 ± 12.12	
	Daughter-in-law	84.85 ± 11.19	
	Son-in-law	92.09 ± 13.40	
	Other	89.56 ± 13.40	
Number of chronic diseases	0	86.37 ± 11.91	1.41 (0.247)
	1	86.07 ± 12.46	
	≥2	88.90 ± 13.62	
Duration of caregiving at home (months) ^†^	0 ^a^	87.70 ± 11.06	4.72 (0.010) b < c
	1–24 ^b^	83.60 ± 14.18	
	≥25 ^c^	88.50 ± 11.92	
Gender of resident in LTCF	Female	87.32 ± 12.45	1.00 (0.321)
	Male	85.73 ± 12.73	
Age of resident in LTCF (years)	≤75	83.71 ± 14.71	2.00 (0.137)
	76–85	86.79 ± 12.34	
	≥86	87.95 ± 11.89	
Size of facility	≤99 beds	90.14 ± 12.52	4.67 (<0.001)
	≥100 beds	83.89 ± 11.78	
Number of visits to the facility (per month) ^†^	≤4 ^a^	88.21 ± 11.61	10.77 (<0.001) a, b > c
	5–9 ^b^	85.36 ± 13.38	
	≥10 ^c^	77.50 ± 15.21	
Duration of each visit (hours) *	≤1	87.86 ± 11.66	7.03 (0.030)
	2–3	83.66 ± 11.57	
	≥4	87.07 ± 24.89	
Number of decision-makers regarding placement of relative in facility	≤1	85.01 ± 11.85	−2.55 (0.011)
	≥2	88.50 ± 12.87	
Family member participation in direct care	Yes	85.77 ± 12.61	−2.96 (0.003)
	No	90.43 ± 11.63	

* Kruskal–Wallis test; ^†^ significant difference between groups based on Scheffe’s post hoc test; ^a, b, c^ Comparison group based on Scheffe’s post hoc test; SD, standard deviation.

**Table 6 ijerph-17-03298-t006:** Correlations between measured variables, *N* = 330.

Variables	1	1-1	1-2	1-3	1-4	2	3	3-1	3-2	3-3
	r (*p*)	r (*p*)	r (*p*)	r (*p*)	r (*p*)	r (*p*)	r (*p*)	r (*p*)	r (*p*)	r (*p*)
1. Family member’s caregiving stress										
1-1. Conflicts with facility staff *	0.75 (<0.001)									
1-2. Internal conflict *	0.83 (<0.001)	0.56 (<0.001)								
1-3. Guilt *	0.66 (<0.001)	0.36 (<0.001)	0.43 (<0.001)							
1-4. Loss *	0.66 (<0.001)	0.16 (0.004)	0.38 (<0.001)	0.35 (<0.001)						
2. Satisfaction of long-term care facilities	−0.24 (<0.001)	−0.33 (<0.001)	−0.23 (<0.001)	−0.02 (0.730)	−0.06 (0.292)					
3. Family member’s satisfaction of care	−0.44 (<0.001)	−0.50 (<0.001)	−0.39 (<0.001)	−0.13 (0.017)	−0.16 (0.005)	0.62 (<0.001)				
3-1. Consideration by staff ^†^	−0.36 (<0.001)	−0.46 (<0.001)	−0.35 (<0.001)	−0.07 (0.200)	−0.10 (0.079)	0.61 (<0.001)	0.90 (<0.001)			
3-2. Efficient management ^†^	−0.43 (<0.001)	−0.47 (<0.001)	−0.36 (<0.001)	−0.20 (<0.001)	−0.16 (0.003)	0.53 (<0.001)	0.87 (<0.001)	0.68 (<0.001)		
3-3. Physical nursing care ^†^	−0.41 (<0.001)	−0.44 (<0.001)	−0.39 (<0.001)	−0.09 (0.090)	−0.17 (0.002)	0.54 (<0.001)	0.89 (<0.001)	0.75 (<0.001)	0.69 (<0.001)	
3-4. Activities ^†^	−0.11 (0.045)	−0.15 (0.009)	−0.08 (0.139)	−0.03 (0.603)	−0.04 (0.466)	0.20 (<0.001)	0.43 (<0.001)	0.27 (<0.001)	0.25 (<0.001)	0.26 (<0.001)

Note: r = Pearson correlation coefficient; *p* = level of significance; * subdomain of family member’s caregiving stress, ^†^ subdomain of family member’s satisfaction with care.

**Table 7 ijerph-17-03298-t007:** Hierarchical multiple regression analysis with family caregivers’ satisfaction with care services in LTCFs, *N* = 330.

Variables	Model Ⅰ			Model Ⅱ		
	β	T	*p*	β	T	*p*
(Constant)		19.08	<0.001		14.75	<0.001
Gender (men = 1) ^†^	0.01	0.24	0.812	−0.03	−0.62	0.533
Perceived stress	0.08	1.51	0.132	−0.03	−0.76	0.446
Relationship with the resident (adult child = 1) ^†^	0.16	1.32	0.186	0.01	0.05	0.963
Relationship with the resident (daughter-in-law = 1) ^†^	0.05	0.47	0.640	−0.03	−0.34	0.731
Relationship with the resident (son-in-law = 1) ^†^	0.16	1.92	0.055	0.07	1.08	0.282
Relationship with the resident (other = 1) ^†^	0.14	1.49	0.136	0.07	1.04	0.301
Duration of caregiving at home (months)	0.13	2.50	0.013	0.04	0.94	0.347
Size of facility (≥100 beds = 1) ^†^	−0.19	−3.52	0.001	−0.13	−3.21	0.001
Number of visits to the facility (per month)	−0.19	−3.41	0.001	−0.10	−2.27	0.024
Duration of each visit (hours) (2–3 h = 1) ^†^	−0.06	−1.23	0.218	−0.04	−1.06	0.288
Duration of each visit (hours) (>3 h = 1) ^†^	0.01	−0.01	0.992	0.02	0.57	0.567
Number of decision-makers who made the decision to place the relative in a facility	0.11	2.03	0.043	0.09	2.14	0.033
Participation of the family member in direct care (yes = 1) ^†^	−0.12	−2.25	0.025	−0.07	−1.64	0.102
Caregiving stress				−0.30	−7.16	<0.001
Satisfaction with the physical housing environment				0.49	11.51	<0.001
R^2^	0.20			0.54		
Adjusted R^2^	0.17			0.52		
R^2^ change				0.34		
F (*p*)	6.13 (<0.001)			24.26 (<0.001)		
Durbin–Watson	1.888					

^†^ Dummy coded; β = regression coefficient; R^2^ = percentage of explained variance; *p* = level of significance.

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
