# Peer review of "Predictors of Satisfaction with Care Services among Family Members of Older Adult Residents of Long-Term Care Facilities"

_ijerph, 2020, doi:10.3390/ijerph17093298_

Round 1

Reviewer 1 Report

Please see attached document for some suggestions to strengthen this interesting article. 

Author Response

Response to Reviewers’ comment

Dear Reviewer:

We wish to thank you for your thoughtful comments and valuable feedback on the manuscript entitled “Predictors of satisfaction with care services among family members of older adult residents of long-term care facilities.” We would like to resubmit the revised manuscript for publication in the International Journal of Environmental Research and Public Health.

We have tried to revise the manuscript according to your suggestions and rewrote or rephrased sections to improve clarity and added further information to explain details about some vague points. For your convenience, we have used red font for the revisions. Please find the following revisions according to reviewer’s comments.

Further, I believe that this revised paper will be of interest to the readership of the  International Journal of Environmental Research and Public Health. Thank you for your consideration. I look forward to hearing from you.

Sincerely,

Hye-Young Jang

Reviewer 2 Report

This paper investigates satisfaction with care services amongst family members of older adult residents of long-term care facilities. This subject has always been of high interest for both academics and practitioners but, given the current circumstances across the world due to COVID-19, the topic is now more important and relevant than ever before.

From my point of view, there are some points that need to be clarified and reviewed by the authors. My main questions and comments are the following:

  • I recommend the authors incorporate a section only with the theoretical framework. Satisfaction with care services were extensively researched over the last decades, with various systematic reviews encompassing the most relevant studies in the field. The authors referred to previous studies, but they do not inform of the results obtained or do so very briefly. I believe there is a need to significantly improve the theoretical background and to indicate how this paper contribute to the existing literature.

  • Is the convenience sample used in the study representative of the analysed population? Do the authors possess that information? If the sample is representative, then it should be stated in the paper.

  • With regards to the satisfaction with Physical Housing Environment, has anyone before undertaken a study where this measure was used?

  • The discussion section needs to be further elaborated. The authors underpinned their results with several studies but do not give a clear explanation of the results that were obtained. The most important predictors of satisfaction with care services were satisfaction with the physical housing environment (β=.49, p<.001), caregiving stress (β=-.30, p<.001), the facility’s size (β=-.13, p=.001), the number of visits to the facility (β=-.10, p=.024), and the number of decision-makers for entering a facility (β=.09, p=.033). Authors should analyse whether the sign is positive or negative and justify their reasoning. It strikes me the negative sign in the number of visits to the facility. What does that mean? Do the number of visits need to be promoted? Do the authors have information about other variables that could be incorporated to the Hierarchical Multiple Regression Analysis, for example, perceived quality of services?

  • What are the limitations of this study?

I hope the authors find these comments useful and you can make good progress on this paper.

Best Regards

Author Response

(The authors gave the same response as above.)

Reviewer 3 Report

Nice paper with very interesting research design.

I have some questions and comments about the text you submitted:

Introduction section: Your introduction focusses a lot on user satisfaction, especially from the perspective of the family. But how is satisfaction related to quality of care? How can services use the information to improve the care. Poor satisfaction does not mean services provide poor quality of care and otherwise. Can you elaborate on that more in the introduction? I would like to see a paragraph on how satisfaction is related to quality of care (more elaborate than it is described now) and why you chose to take the perspective of the family and not of the elderly patient.

Methods:

  • Study design and setting: Please describe the enrollment proces more in detail and place the number of questionnaire in the result section (+ response rate). 
  • How were inappropriate answers defined or detected?
  • Measures: how did you select these measures? If by literature review, please add this information in the methods en results section
  • Statistical analyses: please describe your power calculation here.

Results: 

  • please add some information on response

Discussion:

  • Can you start your discussion with your main findings? 
  • Can you please also add a paragraph on strength and limitations of the study?
  • In the introduction you link satisfaction of users to quality of care and quality improvement. Please elaborate on this in the discussion as well. The results of this paper add more to practice than described now. 

Author Response

(The authors gave the same response as above.)

Round 2

Reviewer 2 Report

Thank you for revising, based on my comments. I now feel comfortable recommending publication to the editor.

Best regards,